# BYTE-LEVEL RECURSIVE CONVOLUTIONAL AUTO-ENCODER FOR TEXT

## ABSTRACT

This article proposes to auto-encode text at byte-level using convolutional networks with a recursive architecture. The motivation is to explore whether it is possible to have scalable and homogeneous text generation at byte-level in a non-sequential fashion through the simple task of auto-encoding. We show that non-sequential text generation from a fixed-length representation is not only possible, but also achieved much better auto-encoding results than recurrent networks. The proposed model is a multi-stage deep convolutional encoder-decoder framework using residual connections (He et al., 2016), containing up to 160 parameterized layers. Each encoder or decoder contains a shared group of modules that consists of either pooling or upsampling layers, making the network recursive in terms of abstraction levels in representation. Results for 6 large-scale paragraph datasets are reported, in 3 languages including Arabic, Chinese and English. Analyses are conducted to study several properties of the proposed model.

## 1 INTRODUCTION

Recently, generating text using convolutional networks (ConvNets) starts to become an alternative to recurrent networks for sequence-to-sequence learning (Gehring et al., 2017). The dominant assumption for both these approaches is that texts are generated one word at a time. Such sequential generation process bears the risk of output or gradient vanishing or exploding problem (Bengio et al., 1994), which limits the length of its generated results. Such limitation in scalability prompts us to explore whether non-sequential text generation is possible.

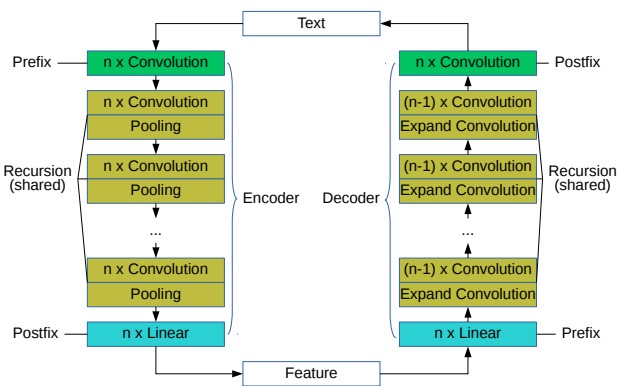

Figure 1: The autoencoder model

Meanwhile, text processing from lower levels than words – such as characters (Zhang et al., 2015) (Kim et al., 2016) and bytes (Gillick et al., 2016) (Zhang & LeCun, 2017) – is also being explored due to its promise in handling distinct languages in the same fashion. In particular, the work by Zhang & LeCun (2017) shows that simple one-hot encoding on bytes could give the best results for text classification in a variety of languages. The reason is that it achieved the best balance between computational performance and classification accuracy. Inspired by these results, this article explores auto-encoding for text using byte-level convolutional networks that has a recursive structure, as a first step towards low-level and non-sequential text generation.

For the task of text auto-encoding, we should avoid the use of common attention mechanisms like those used in machine translation Bahdanau et al. (2015), because they always provide a direct information path that enables the auto-encoder to directly copy from the input. This diminishes the

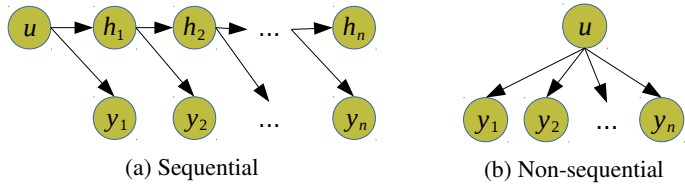

Figure 2: Sequential and non-sequential decoders illustrated in graphical models. $u$ is a vector containing encoded representation. $y_i$'s are output entities. $h_i$'s are hidden representations. Note that they both imply conditional independence between outputs conditioned by the representation $u$.

purpose of studying the representational ability of different models. Therefore, all models considered in this article would encode to and decode from a fixed-length vector representation.

The paper by Zhang et al. (2017) was an anterior result on using word-level convolutional networks for text auto-encoding. This article differs from it in several key ways of using convolutional networks. First of all, our models work from the level of bytes instead of words, which arguably makes the problem more challenging. Secondly, our network is dynamic with a recursive structure that scales with the length of input text, which by design could avoid trivial solutions for auto-encoding such as the identity function. Thirdly, by using the latest design heuristics such as residual connections (He et al., 2016), our network can scale up to several hundred of layers deep, compared to a static network that contains a few layers.

In this article, several properties of the auto-encoding model are studied. The following is a list.

1. Applying the model to 3 languages – Arabic, Chinese and English – shows that the model can handle all different languages in the same fashion with equally good accuracy.

2. Comparisons with long short-term memory (LSTM) (Hochreiter & Schmidhuber, 1997) show a significant advantage of using convolutional networks for text auto-encoding.

3. We determined that a recursive convolutional decoder like ours can accurately produce the end-of-string byte, despite that the decoding process is non-sequential.

4. By studying the auto-encoding error when the samples contain randomized noisy bytes, we show that the model does not degenerate to the identity function. However, it can neither denoise the input very well.

5. The recursive structure requires a pooling layer. We compared between average pooling, L2 pooling and max-pooling, and determined that max-pooling is the best choice.

6. The advantage of recursion is established by comparison against a static model that does not have shared module groups. This shows that linguistic heuristics such as recursion is useful for designing models for language processing.

7. We also explored models of different sizes by varying the maximum network depth from 40 to 320. The results show that deeper models give better results.

## 2 BYTE-LEVEL RECURSIVE CONVOLUTIONAL AUTO-ENCODER

In this section, we introduce the design of the convolutional auto-encoder model with a recursive structure. The model consists of 6 groups of modules, with 3 for the encoder and 3 for the decoder. The model first encodes a variable-length input into a fixed-length vector of size 1024, then decodes back to the same input length. The decoder architecture is a reverse mirror of the encoder. All convolutional layers in this article have zero-padding added to ensure that each convolutional layer outputs the same length as the input. They also all have feature size 256 and kernel size 3. All parameterized layers in our model use ReLU (Nair & Hinton, 2010) as the non-linearity.

In the encoder, the first group of modules consist of $n$ temporal (1-D) convolutional layers. It accepts an one-hot encoded sequence of bytes as input, where each byte is encoded as a 256-dimension vector. This first group of modules transforms the input into an internal representation. We call this group of modules the prefix group. The second group of modules consists of $n$ temporal convolutional layers plus one max-pooling layer of size 2. This group reduces the length of input by a factor

of 2, and it can be applied again and again to recursively reduce the representation length. Therefore, we name this second group the recursion group. The recursion group is applied until the size of representation becomes 1024, which is actually a feature of dimension 256 and length 4. Then, following the final recursion group is a postfix group of $n$ linear layers for feature transformation.

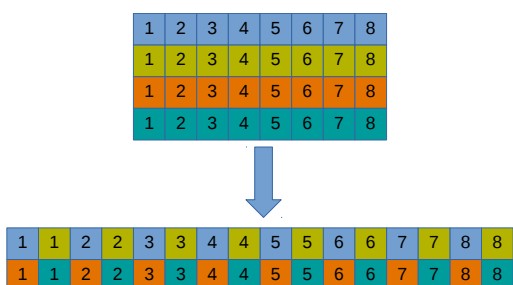

The decoder is a symmetric reverse mirror of the encoder. The decoder prefix group consists of $n$ linear layers, followed by a decoder recursion group that expand the length of representation by a factor of 2. This expansion is done at the first convolutional layer of this group, where it outputs 512 features that will be reshaped into 256 features. The reshaping process we use ensures that feature values correspond to nearby field of view in the input, which is similar to the idea of sub-pixel convolution (or pixel shuffling) (Shi et al., 2016). Figure 3 depics this reshaping process for transforming representation of feature size 4 and length 8 to feature size 2 and length 16. After this recursion group is applied several times (same as that of the encoder recursion group), a decoder postfix group of $n$ convolutional layers is applied to decode the recursive features into a byte sequence.

Figure 3: The reshaping process. This demonstrates the reshaping process for transforming a representation of feature size 4 and length 8 to feature size 2 and length 16. Differen colors represent different source features, and the numbers are indices in length dimension.

The final output of the decoder is interpreted as probabilities of bytes after passing through a softmax function. Therefore, the loss we use is simply negative-log likelihood on the individual softmax outputs. It is worth noting that this does not imply that the output bytes are unconditionally independent of each other. For our non-sequential text decoder, the independence between output bytes is conditioned on the representation from the encoder, meaning that their mutual dependence is modeled by the decoder itself. Figure 2 illustrates the difference between sequential and non-sequential text generation using graphical models.

Depending on the length of input and size of the encoded representation, our model can be extremely deep. For example, with $n = 8$ and encoding dimension 1024 (reduced to a length 4 with 256 features), for a sample length of 1024 bytes, the entire model has 160 parameterized layers. Training such a deep dynamic model can be very challenging using stochastic gradient descent (SGD) due to the gradient vanishing problem (Bengio et al., 1994). Therefore, we use the recently proposed idea of residual connections (He et al., 2016) to make optimization easier. For every pair of adjacent parameterized layers, the input feature representation is passed through to the output by addition. We were unable to train a model designed in this fashion without such residual connections.

For all of our models, we use an encoded representation of dimension 1024 (recursed to length of 4 with 256 features). For an input sample of arbitrary length $l$, we first append the end-of-sequence null byte to it, and then pad it to length $2^{\lceil \log_2(l+1) \rceil}$ with all zero vectors. This makes the input length a base-2 exponential of some integer, since the recursion groups in both encoder and decoder either reduce or expand the length of representation by a factor of 2. If $l < 4$, it is padded to size of 4 and does not pass through the recursion groups. It is easy to see that the depth of this dynamic network for a sample of length $l$ is on the order of $\log_2 l$, potentially making the hidden representations more efficient and easier to learn than recurrent networks which has a linear order in depth.

## 3 RESULT FOR MULTI-LINGUAL AUTO-ENCODING

In this section, we show the results of our byte-level recursive convolutional auto-encoder.

### 3.1 DATASET

All of our datasets are at the level of paragraphs. Minimal pre-processing is applied to them since our model can be applied to all languages in the same fashion. We also constructed a dataset with samples mixed in all three languages to test the model's ability to handle multi-lingual data.

Table 1: Datasets

| NAME | ARTICLE | | PARAGRAPH | | LANGUAGE |
|---|---|---|---|---|---|
| | TRAIN | TEST | TRAIN | TEST | |
| enwiki | 7,634,438 | 850,457 | 41,256,261 | 4,583,893 | English |
| hudong | 1,618,817 | 180,278 | 53,675,117 | 5,999,920 | Chinese |
| argiga | 3,011,403 | 334,764 | 27,989,646 | 3,116,719 | Arabic |
| engiga | 8,887,583 | 988,513 | 116,456,520 | 12,969,170 | English |
| zhgiga | 5,097,198 | 567,179 | 38,094,390 | 4,237,643 | Chinese |
| allgiga | 16,996,184 | 1,89,0456 | 182,540,556 | 20,323,532 | Multi-lingual |

**enwiki.** This dataset contains paragraphs from the English Wikipedia [1], constructed from the dump on June 1st, 2016. We were able to obtain 8,484,895 articles, and then split our 7,634,438 for training and 850,457 for testing. The number of paragraphs for training and testing are therefore 41,256,261 and 4,583,893 respectively.

**hudong.** This dataset contains pragraphs from the Chinese encyclopedia website `baike.com` [2]. We crawled 1,799,095 article entries from it and used 1,618,817 for training and 180,278 for testing. The number of paragraphs for training and testing are 53,675,117 and 5,999,920.

**argiga.** This dataset contains paragraphs from the Arabic Gigaword Fifth Edition release (Parker et al., 2011a), which is a collection of Arabic newswire articles. In total there are 3,346,167 articles, and we use 3,011,403 for training and 334,764 for testing. As a result, we have 27,989,646 paragraphs for training and 3,116,719 for testing.

Table 2: Training and testing byte-level errors

| DATASET | LANGUAGE | TRAIN | TEST |
|---|---|---|---|
| enwiki | English | 3.34% | 3.34% |
| hudong | Chinese | 3.21% | 3.16% |
| argiga | Arabic | 3.08% | 3.09% |
| engiga | English | 2.09% | 2.08% |
| zhgiga | Chinese | 5.11% | 5.24% |
| allgiga | Multi-lingual | 2.48% | 2.50% |

**engiga.** This dataset contains paragraphs from the English Gigaword Fifth Edition release (Parker et al., 2011c), which is a collection of English newswire articles. In total there are 9,876,096 articles, and we use 8,887,583 for training and 988,513 for testing. As a result, we have 116,456,520 paragraphs for training and 12,969,170 for testing.

**zhgiga.** This dataset contains paragraphs from the Chinese Gigaword Fifth Edition release (Parker et al., 2011b), which is a collection of Chinese newswire articles. In total there are 5,664,377 articles, and we use 5,097,198 for training and 567,179 for testing. As a result, we have 38,094,390 paragraphs for training and 4,237,643 for testing.

**allgiga.** Since the three Gigaword datasets are very similar to each other, we combined them to form a multi-lingual dataset of newswire article paragraphs. In this dataset, there are 18,886,640 articles with 16,996,184 for training and 1,890,456 for testing. The number of paragraphs for training and testing are 182,540,556 and 20,323,532 respectively.

Table 1 is a summary of these datasets. For such large datasets, testing time could be unacceptably long. Therefore, we report all the results based on 1,000,000 samples randomly sampled from either training or testing subsets depending on the scenario. Very little overfitting was observed even for our largest model.

## 3.2 RESULT

---

[1] `https://en.wikipedia.org`
[2] `http://www.baike.com/`

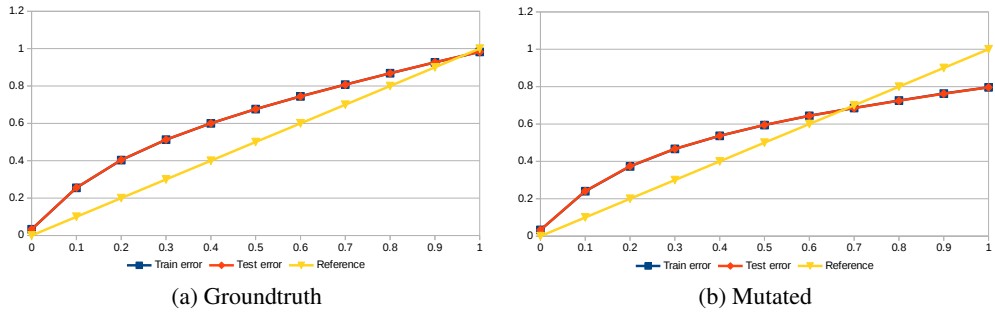

(a) Groundtruth                                        (b) Mutated

Figure 5: Byte-level errors with respect to randomly mutated samples

Regardless of dataset, all of our text auto-encoders are trained with the same hyper-parameters using stochastic gradient descent (SGD) with momentum (Polyak, 1964) (Sutskever et al., 2013). The model we used has $n = 8$ – that is, there are 8 parameterized layers in each of prefix, recursion and postfix module groups, for both the encoder and decoder. Each training epoch contains 1,000,000 steps, and each step is trained on a randomly selected sample with length up to 1024 bytes. Therefore, the maximum model depth is 160.

Table 3: Byte-level errors for long short-term memory (LSTM) recurrent network

| DATASET | LANGUAGE | TRAIN | TEST |
|---------|----------|-------|------|
| enwiki | English | 67.71% | 67.80% |
| fg hudong | Chinese | 64.47% | 64.56% |
| argiga | Arabic | 61.23% | 61.29% |
| engiga | English | 70.47% | 70.45% |
| zhgiga | Chinese | 75.91% | 75.90% |
| allgiga | Multi-lingual | 72.39% | 72.44% |

We only back-propagate through valid bytes in the output. Note that each sample contains a end-of-sequence byte ("null" byte) by design.

We set the initial learning rate to 0.001, and half it every 10 epoches. A momentum of 0.9 is applied to speed up training. A small weight decay of 0.00001 is used to stabilize training. Depending on the length of each sample, the encoder or decoder recursion groups are applied for a certain number of times. We find that dividing the gradients of these recursion groups by the number of shared clones can speed up training. The trainng process stops at the 100th epoch.

Note that because engiga and allgiga datasets have more than 100,000,000 training samples, when training stops the model has not seen the entirety of training data. However, further training does not achieve any observable improvement. Table 2 details the byte-level errors for our model on all of the aforementioned datasets. These results indicate that our models can achieve very good error rates for auto-encoding in different languages. The result for allgiga dataset also indicates that the model has no trouble in learning from multi-lingual datasets that contains samples of very different languages.

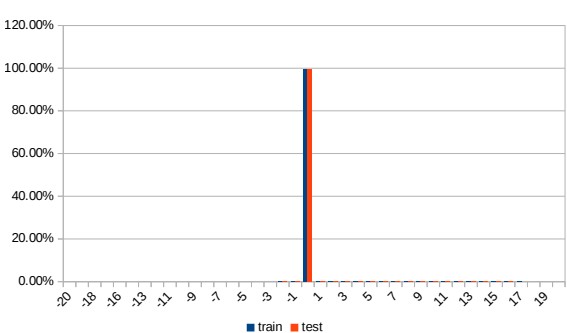

Figure 4: The histogram of length difference

## 4   DISCUSSION

This section offers comparisons with recurrent networks, and studies on a set of different properties of our proposed auto-encoding model. Most of these results are performed using the enwiki dataset.

## 4.1 COMPARISON WITH RECURRENT NETWORKS

We constructed a simple baseline recurrent network using the "vanilla" long short-term memory units (Hochreiter & Schmidhuber, 1997). In this model, both input and output bytes are embedded into vectors of dimension 1024 so that we can use a hidden representation of dimension 1024. The encoder reads the text in reverse order, which was observed by Sutskever et al. (2014) reversing the input sequence can improve quality of outputs. The 1024-dimension hidden output of the last cell is used as the input for the decoder.

The decoder also and input and output bytes embedded into vectors of dimension 1024 and use a hidden representation of dimension 1024. During decoding, the most recently generated byte is fed to the next time step. This is called "teacher forcing" which is observed to improve the auto-encoding result in our case. The decoding process uses a beam search algorithm of size 2. During learning, we only back-propagate through the most likely sequence after beam search.

Table 3 details the result for LSTM. The byte-level errors are so large that the results of our models in table 2 are at least one magnitude of doing better. The fundamental limitation of recurrent networks is that regardless of the level of entity (word, character or byte), they can remember around up to 50 of them accurately, and then failed to accurate predict them afterwards. By construction our recursive non-sequential text generation process could hopefully be an alternative solution for this, as already evident in the results here.

## 4.2 END OF SEQUENCE

One thing that makes a difference between sequential and non-sequential text generation is how to decide when to end the generated string of bytes. For sequential generative process such as recurrent decoders, we could stop when some end-of-sequence symbol is generated. For non-sequential generative process, we could regard the first encountered end-of-sequence symbol as the mark for end, despite that it will inevitably generate some extra symbols after it. Then, a natural question to ask it, is this simple way of determining end-of-sequence effective?

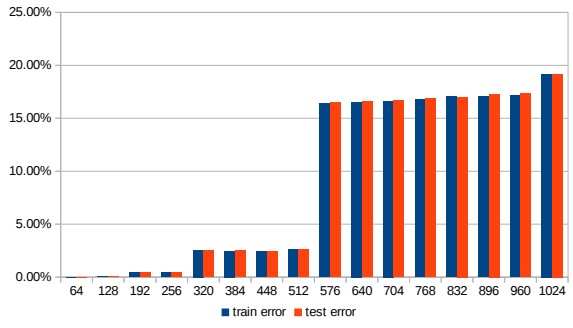

Figure 6: Byte-level error by length

To answer this question, we computed the difference of end-of-sequence symbols between generated text and its groundtruth for 1,000,000 samples, for both the training and testing subsets of the enwiki dataset. What we discovered is that the distribution of length difference is highly concentrated at 0, at 99.63% for both training and testing. Figure 4 shows the full histogram, in which length differences other than 0 is barely visible. This suggests that our non-sequential text generation process can model the end-of-sequence position pretty accurately. One reason for this is that for every samples we have an end-of-sequence symbol – the "null" byte – such that the network has learned to model it pretty early on during the training process.

## 4.3 RANDOM PERMUTATION OF SAMPLES

One potential problem specific to the task if auto-encoding is the risk of learning the degenerate solution – the identity function. One way to test this is to mutate the input bytes randomly and see whether the error rates match with the mutation probability. We experimented with mutation probability from 0 to 1 with an interval of 0.1, and for each case we tested the byte-level errors for 100,000 samples in both training and testing subsets of the enwiki dataset.

Note that we can compute the byte-level errors in 2 ways. The first is to compute the errors with respect to the groundtruth samples. If the solution is degenerated to the identity function, then the byte-level errors should correlate with the probability of mutation. The second is to compute the errors with respect to the mutated samples. If the solution is degenerated to the identity function,

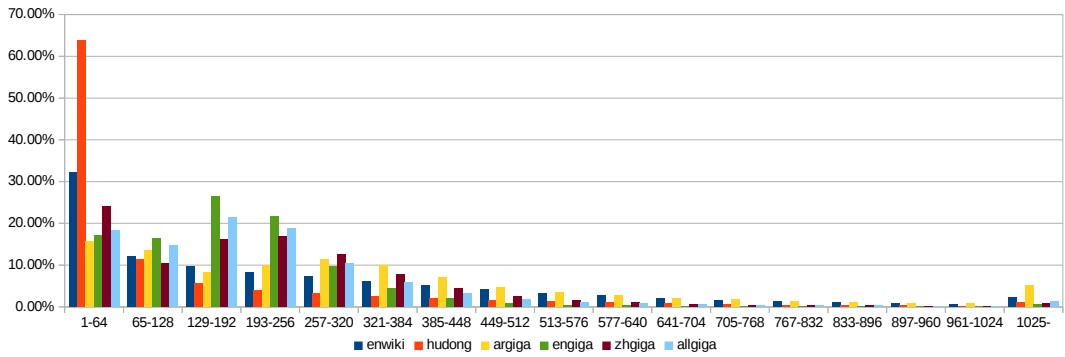

Figure 7: Histogram of sample frequencies in different lengths

then the byte-level errors should be near 0 regardless of the mutation probability. Figure 5 shows the results in these 2 ways of computing errors, and the result strongly indicates that the model has not degenerated to learning the identity function.

It is worth noting that the errors with respect to the groundtruth samples in figure 5 also demonstrate that our model lacks the ability to denoise mutated samples. This can be seen from the phenomenon that the errors for each mutation probability is higher than the reference diagonal value, instead of lower. This is due to the lack of a denoising criterion in our training process.

Table 4: Byte-level errors for different pooling layers

| POOL | TRAIN | TEST |
|---|---|---|
| max | 3.34% | 3.34% |
| average | 7.91% | 7.98% |
| L2 | 6.85% | 6.77% |

### 4.4 SAMPLE LENGTH

We also conducted experiments to show how does the byte-level errors vary with respect to the sample length. Figure 7 shows the histogram of sample lengths for all datasets. It indicates that a majority of paragraph samples can be well modeled under 1024 bytes. Figure 6 shows the byte-level error of our models with respect to the length of samples. This figure is produced by testing 1,000,000 samples from each of training and testing subsets of enwiki dataset. Each bin in the histogram represent a range of 64 with the indicated upper limit. For example, the error at 512 indicate errors aggregated for samples of length 449 to 512.

One interesting phenomenon is that the errors are highly correlated with the number of recursion groups applied for both the encoder and the decoder. In the plot, bins 64, 128, 192-256, 320-512, 576-1024 represent recursion levels of 4, 5, 6, 7, 8 respectively. The errors for the same recursion level are almost the same to each other, despite huge length differences when the recursion levels get deep. The reason for this is also related to the fact that there there tend to be more shorter texts than longer ones in the dataset, as evidenced in figure 7.

Table 5: Byte-level errors for recursive and static models

| MODEL | TRAIN | TEST |
|---|---|---|
| recursive | 3.34% | 3.34% |
| static | 8.01% | 8.05% |

### 4.5 POOLING LAYERS

This section details an experiment in studying how do the training and testing errors vary with the choice of pooling layers in the encoder network. The experiments are conducted on the aforementioned model with $n = 8$, and replacing the max-pooling layer in the encoder with average-pooling or L2-pooling layers. Table 4 details the result. The numbers strongly indicate that max-pooling is the best choice. Max-pooling selects the largest values in its field of view, helping the network to achieve better optima (Boureau et al., 2010).

### 4.6 RECURSION

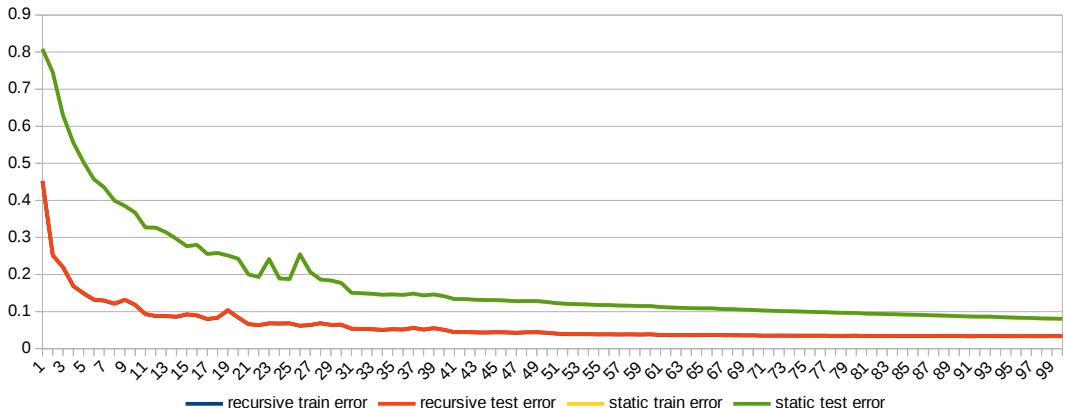

Figure 8: Errors during training for recursive and static models.

The use of recursion in the proposed model is from a linguistic intuition that the structure may help the model to learn better representations. However, there is to guarantee that such intuition could be helpful for the model unless comparison is done with a static model that takes fixed-length inputs and pass through a network with the same architecture of the recursion groups without weight sharing.

Figure 8 shows the training and testing errors when training a static model with the same hyper-parameters. The static model takes 1024 bytes, and zero vectors are padded if the sample length is smaller. The recursion group is therefore applied for 8 times in both the encoder and decoder, albeit their weights are not shared. The result indicates that a recursive model not only learns faster, but can also achieve better results. Table 5 lists the byte-level errors.

Table 6: Byte-level errors depending on model depth

| $n$ | DEPTH | TRAIN | TEST |
|---|---|---|---|
| 2 | 40 | 9.05% | 9.07% |
| 4 | 80 | 5.07% | 5.11% |
| 8 | 160 | 3.34% | 3.34% |
| 16 | 320 | 2.91% | 2.92% |

### 4.7 MODEL DEPTH

This section explores whether varying the model size can make a different on the result. Table 6 lists the training and testing errors of different model depths with $n \in \{2, 4, 8, 16\}$. The result indicates that best error rates are achieved with the largest model, with very little overfitting. This is partly due to the fact that our datasets are quite large for the models in question.

## 5 CONCLUSION

In this article, we propose to auto-encode text using a recursive convolutional network. The model contains 6 parts – 3 for the encoder and 3 for the decoder. The encoder and decoder both contain a prefix module group and a postfix module group for feature transformation. A recusion module group is included in between the prefix and postfix for each of the encoder and decoder, which recursively shrink or expand the length of representation. As a result, our model essentially generate text in a non-sequential fashion.

Experiments using this model are done on 6 large scale datasets in Arabic, Chinese and English. Comparison with recurrent networks is offered to show that our model achieved great results in text auto-encoding. Properties of the proposed model are studied, including its ability to produce the end-of-sequence symbol, whether the model degenerates to the identity function, and variations of pooling layers, recursion and depth of models. In the future, we hope to extend our models to non-sequential generative models without inputs, and use it for more sequence-to-sequence tasks such as machine translation.

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
