# OpenReview forum: "Byte-Level Recursive Convolutional Auto-Encoder for Text"
_ICLR.cc/2018/Conference — Reject_

### Official Review · AnonReviewer2 · 2017-11-28
**Minor step forward, but good investigation & analysis**

**Rating:** 7
**Confidence:** 4

**Review:**


This paper presents a convolutional auto-encoder architecture for text encoding and generation. It works on the character level and contains a recursive structure which scales with the length of the input text. Building on the recent state-of-the-art in terms of architectural components, the paper shows the feasibility of this architecture and compares it to LSTM, showing the cnn superiority for auto-encoding.

The authors have decided to encode the text into a length of 1024 - Why? Would different lengths result in a better performance?

You write "Minimal pre-processing is applied to them since our model can be applied to all languages in the same fashion." Please be more specific. Which pre-processing do you apply for each dataset?

I wonder if the comparison to a simple LSTM network is fair. It would be better to use a 2- or 3-layer network. Also, BLSTM are used nowadays.

A strong part of this paper is the large amount of investigation and extra experiments.
Minor issues:
Please correct minor linguistic mistakes as well as spelling mistakes. In Fig. 3, for example, the t of Different is missing.

An issue making it hard to read the paper is that most of the figures appear on another page than where they are mentioned in the text.

the authors have chosen to cite a work from 1994 for the vanishing gradient problem. Note, that many (also earlier) works have reported this problem in different ways. A good analysis of all researches is performed in Hochreiter, S., Bengio, Y., Frasconi, P., and Schmidhuber, J. (2001) "Gradient flow in recurrent nets: the difficulty of learning long-term dependencies".

---

### Official Review · AnonReviewer3 · 2017-11-30
**Related work and other tasks for experiments.**

**Rating:** 5
**Confidence:** 3

**Review:**

The authors propose autoencoding text using a byte-level encoding and a convolutional network with shared filters such that the encoder and decoder should exhibit recursive structure. They show that the model can handle various languages and run various experiments testing the ability of the autoencoder to reconstruct the text with varying lengths, perturbations, depths, etc.

The writing is fairly clear, though many of the charts and tables are hard to decipher without labels (and in Figure 8, training errors are not visible -- maybe they overlap completely?).

Main concern would be the lack of experiments showing that the network learns meaningful representations in the hidden layer. E.g. through semi-supervised learning experiments or experiments on learning semantic relatedness of sentences. Obvious citations such as https://arxiv.org/pdf/1511.06349.pdf and https://arxiv.org/pdf/1503.00075.pdf are missing, along with associated baselines. Although the experiment with randomly permuting the samples is nice, would hesitate to draw any conclusions without results on downstream tasks and a clearer survey of the literature.

---

### Official Review · AnonReviewer1 · 2017-12-05
**This paper proposes to encode non-sequential text at the byte level using a convolutional auto encoder  as an alternative to recurrent architectures. The proposed network offers scalability, operates at the byte-level using a fixed length one-hot encoding vector as input, and  includes a recursive structure to prevent trivial encoding. Results are presented on three languages and compared to other architectures.**

**Rating:** 5
**Confidence:** 5

**Review:**

The paper aims to illustrated the representation learning ability of the convolutional autoencoder with residual connections is  proposed by to encode text at the byte level.  The authors apply the proposed architecture to 3 languages and run comparisons with an LSTM.  Experimental results  with different perturbation of samples, pooling layers, and sample lengths are presented.

The writing is fairly clear, however the presentation of tables and figures could be done better, for example, Fig. 2 is referred  to in page 3,  Table 2 which contains results is referred to on page 5, Fig 4 is referred to in page 6 and appears in page 5, etc.

What kind of minimal preprocessing is done on the text? Are punctuations removed? Is casing retained? How is the space character encoded?

Why was the encoded dimension always fixed at 1024?  What is the definition of a sample here?

The description of the various data sets could be moved to a table/Appendix, particularly since most of the results are presented on the enwiki dataset, which would lead to better readability of the paper.  Also results are presented only on a random 1M sample selected from these data sets, so the need for this whole page goes away.

Comparing Table 2 and Table 3, the LSTM is at 67% error on the test set while the proposed convolutional autoencoder is at 3.34%.  Are these numbers on the same test set?    While the argument that the LSTM does not generalize well due to the inherent memory learnt is reasonable, the differences in performance cannot be explained away with this. Can you please clarify this further?

It appears that the byte error shoot up for sequences of length 512+ (fig. 6 and fig. 7) and seems entirely correlated with the amount of data than recursion levels.

How do you expect these results to change for a different subset selection of training and test samples? Will Fig. 7 and Fig. 6 still hold?

In Fig, 8, unless the static train and test error are exactly on top of the  recursive errors, they are not visible.  What is the x-axis in Fig. 8?  Please also label axes on all figures.

While the datasets are large and would take a lot of time to process for each case study, a final result on the complete data set, to illustrate if the model does learn well with lots of data would have been useful.  A table showing generated sample text would also clarify the power of the model.

With the results presented,  with a single parameter setting, its hard to determine what exactly the model learns and why.

---

### Public Comment · ~Xiang_Zhang1 · 2018-01-30
**Thanks for the review!**

We want to express our thanks to the reviewers' and committee's suggestions. They are very useful and helpful for us to understand how the community understands our work. We are working on incorporating some of the suggested experiments to a next version of our paper. However, these experiments took longer than expected and could not be published before the rebuttal or paper decision deadline. Hopefully we will be able to provide a revised version soon for another venue. We will update here again when it is done. Thanks!

---

### Decision · Program_Chairs · 2018-01-29
**ICLR 2018 Conference Acceptance Decision**

**Decision:**

Reject

**Comment:**

This paper presents a method for using byte level convolutional networks for building text-based autoencoders.  They show that these models do well compared to RNN-based methods which model text in a sequence.  Evaluation is solely based on byte level prediction error.   The committee feels that the paper would have been stronger if evaluation was on some actual task (say summarization, Miao and Blunsom, for example) and show that it works as well as RNNs, the paper would have been stronger.